# Plant Nitrogen and Phosphorus Resorption in Response to Varied Legume Proportions in a Restored Grassland

**DOI:** 10.3390/plants9030292

**Published:** 2020-03-01

**Authors:** Qiang Li, Daowei Zhou, Matthew D. Denton

**Affiliations:** 1Northeast Institute of Geography and Agroecology, Chinese Academy of Sciences, Changchun 130102, China; zhoudaowei@neigae.ac.cn; 2Jilin Provincial Key Laboratory of Grassland Farming, Northeast Institute of Geography and Agoecology, Chinese Academy of Sciences, Changchun 130102, China; 3School of Agriculture, Food and Wine, The University of Adelaide, Waite Campus, Urrbrae, SA 5064, Australia; matthew.denton@adelaide.edu.au

**Keywords:** nutrient utilization, nutrient limiting, litter, biological nitrogen fixation, legume–grass mixture

## Abstract

An in-depth assessment of plant nutrient resorption can offer insights into understanding ecological processes and functional responses to biotic and abiotic changes in the environment. The legume proportion in a mixed grassland can drive changes in the soil environment and plant relationships, but little information is available regarding how the legume proportion influences plant nutrient resorption in mixed grasslands. In this study, three mixed communities of *Leymus chinensis (Trin.) Tzvel.* and *Medicago sativa* L. differing in legume proportion (Low-L, with 25% legume composition; Mid-L, with 50% legume composition; High-L, with 75% legume composition) were established with four replicates in a degraded grassland. Four years after establishing the mixed grassland, the quantity of biological N_2_ fixation by *M. sativa*, the availabilities of water and nitrogen (N) and phosphorus (P) in soil were examined, and the concentrations and resorption of leaf N and P for both species were measured during forage maturation and senescence. The results showed Mid-L had greater biological N_2_ fixation and soil N availability than Low-L and High-L, while the High-L had lower soil water and P availability, but a greater soil available N:P ratio compared with Low-L and Mid-L. Legume proportion did not alter N or P concentrations of mature leaves. However, in Mid-L N resorption was reduced by 8 to 16% for the two mixed-species compared with Low-L and High-L. High-L enhanced P resorption by 20 to 24% in both plant species compared with Low-L. The *L. chinensis* and *M. sativa* responded differently to varied legume proportion in terms of P resorption. It was concluded that legume proportion drove changes in soil nutrient availability of mixed communities, which primarily altered plant nutrient resorption during senescence, but had no influence on the nutrient concentrations of mature plants. A moderate legume proportion reduced N resorption, and increased senesced leaf N concentration of grass and legume species. The difference in P resorption by two mixed-species significantly changed the interspecific difference of senesced leaf P concentration and the N:P ratio with varied legume proportion.

## 1. Introduction

Nutrient resorption is defined as the translocation of nutrients from senescing plant tissues to growing plant tissues [1], which potentially increases the nutrient use efficiency of the plant [2], and reduces growth limitation by soil and fertilizer [3,4]. As a consequence, nutrient resorption is a plant adaptive mechanism for nutrient-limited conditions [5]. In addition, nutrient resorption regulates key ecosystem processes such as community assembly, biomass production and litter decomposition by altering the inter-specific relationships [6,7], carbon accumulation with nutrient acquisition [8], and litter quality [9,10]. Thus, an in-depth assessment of the influence of biotic and abiotic environments on nutrient resorption would improve our understanding of ecological processes and functional response to environmental changes.

Nutrient resorption efficiency (NuRE), the percentage of nutrients that are re-absorbed, is generally used to quantify plant nutrient resorption [11,12]. Globally averaged, senesced leaves can resorb 62% of nitrogen (N) and 65% of phosphorus (P), respectively [10]. Generally, nutrient resorption efficiencies are reduced with increased nutrient availability in the environment [4,13]. However, this pattern can be affected by climate, soil stoichiometry, ecosystem type and plant species. Increasing water availability can alter plant N resorption response to N enrichment [14], and ecosystems and plant species differentially respond to nutrient enrichment in terms of nutrient resorption [4,15]. N has a key role in regulating plant growth and physiology [16,17], which demonstrates its importance in modifying plant nutrient resorption [5,18]. The responses of plant N and P resorption to varying N availability have been previously examined [13,14,15,16,19]. However, nutrient resorption responses to increasing N availability can vary from no response [19], to negative [13] or positive responses [15,20]. The inconsistent conclusions mentioned above may result from differences in data collection and analytical methods (i.e., using nutrient pools or nutrient concentrations to calculate nutrient resorption efficiencies) [4], but might be due to climate, plant species, soil nutrient backgrounds, and their interactions. Thus, further research is necessary to analyze the factors regulating plant nutrients resorption with changes in N availability.

Legumes supply N via symbiotic fixation of atmospheric N [21,22]. Fixed N not only supports the legume N requirements, but can be transferred into soil and to coexisting plants via root exudation, and decomposition of legume residues [23,24,25], increasing N availability for coexisting non-legumes. Thus, legumes have been widely introduced into grasslands to improve forage production and soil fertility [26,27,28]. Since the total N fixation by legumes is dependent on legume abundance in the grassland [29], increasing the proportion of legume may enhance plant N availability in soil. The changes in legume proportion also directly alter the community structure of grassland, including species richness and dominance. Variation in grassland community structure will likely modify the plant nutrient resorption responses to legume-driven N enrichment. Recent research has suggested that species richness mediated the within-species nutrient resorption in grassland ecosystems [30]. However, limited research has focused on understanding the effects on plant nutrient resorption, and, in particular, the influence of the proportion of legumes in pastures. It remains uncertain whether increasing the legume proportion can improve N availability to the legume itself, because a greater legume proportion in pastures can reduce N fixation efficiencies of legumes [28,31]. In addition, the legume proportion may influence the other environment variables such as soil water [32], and soil P availability [33]. Increasing the legume proportion may intensify the water and P limitation in a grassland ecosystem, due to increased water and P requirements to support legume growth [34,35]. However, other studies reported that legume pasture mixtures or crops could increase plant P availability in the soil through rhizosphere P mobilization [36,37]. Therefore, there is great uncertainty about how available plant nutrients change following legumes, and if these changes alter plant nutrient resorption responses to varied legume proportions in grasslands. Moreover, nutrient utilization differences likely demonstrate inter-specific variation in plant nutrient resorption in response to changing environments [10,38]. Previous research indicated that N-fixing legumes generally had a lower N resorption efficiency and greater P resorption efficiency compared with non-legume plants [10]. However, an unresolved question is how differently legume and non-legume plants respond to environmental changes driven by legume proportions in terms of nutrient resorption.

In the present study, changes in soil water and nutrient availability driven by legume proportion were examined, and their influences on plant nutrient concentrations and nutrient resorption by legumes and coexisting grasses were assessed in a restored grassland. We focused particularly on demonstrating the inter-species differences of grass and legume species, in terms of nutrient resorption responses and their regulating factors with legume proportions in mixed grasslands. 

## 2. Results

### 2.1. Precipitation during Growing Season

From May to September (growing season) in 2010, the total precipitation was 317 mm. No rainfall event occurred within 15 days near our soil sampling date (Figure 1).

### 2.2. Soil Moisture and [N, P] Availability

The High-L led to the significantly lower soil water content at the 0–40 cm soil depth compared with other legume proportions (Figure 2a). Soil inorganic N concentration was significantly greater under Mid-L (Figure 2b). Soil available P concentration decreased under high-L (Figure 2c). The Mid-L and High-L induced greater plant-available soil N:P compared with Low-L (Figure 2d). 

### 2.3. Plant Density and Biomass

Legume sowing proportion had no significant effect on total plant density of grass and alfalfa, while Mid-L and High-L had greater total above-ground biomass of grass and alfalfa compared with Low-L (Figure 3a,c). For Low-L, Mid-L and High-L, the observed percentage of legume densities was 25.2%, 48.2% and 73.9%, respectively in 2010 (Figure 3b). Increasing legume proportion significantly enhanced the root biomass of mixed grassland (Figure 3d).

### 2.4. N Transfer in L. Chinensis and Biological N Fixation of M. Sativa 

The Mid-L induced a significantly higher N transfer (%N_trans_) in *Leymus chinensis* shoots compared with other mixtures (Figure 4a). The High-L had a lower %Ndfa for *Medicago sativa*, while Mid-L had greater total biological N fixation (Figure 4b) [39]. The %Ndfa of *M. sativa* was positively correlated to soil water content (Figure 4c).

### 2.5. Leaf [N, P] concentrations and [N, P] resorption 

Legume proportion did not alter the N and P concentrations and N:P ratios of green leaves for *M. sativa* and *L. chinensis* (Table 1; Figure 5a,b,e). Among the three mixtures, the senesced leaf N concentrations of *M. sativa* and *L. chinensis* were significantly greater under Mid-L (Figure 5c). Increasing legume proportion significantly reduced senesced leaf P concentration of *M. sativa* and *L. chinensis* (Figure 5d), and regulated the inter-species difference of senesced leaf N:P ratios (Figure 5f). Regardless of legume proportion, *L. chinensis* had greater N and P resorption efficiencies compared to *M. sativa* (Table 1; Figure 6a,b). The N resorption efficiencies (NRE) of *L. chinensis* and *M. sativa* were significantly lower under Mid-L (Figure 6a). In general, increasing legume proportion enhanced P resorption efficiency (PRE) of *M. sativa* and *L. chinensis.* However, the PRE of *L. chinensis* and *M. sativa* responded differently to varied legume proportions (Table 1; Figure 6b).

### 2.6. The Correlations between Nutrients Resorption and Legume-Driven N, and Environmental Factors

Senesced leaf N concentration was positively correlated to %Ndfa (Figure 7a), while NRE had weakly negative correlation with %Ndfa for *M. sativa* (Figure 7c); Senesced leaf N concentration was positively correlated with %N_trans_ values for *L. chinensis* (Figure 7b), while NRE was negatively correlated with %N_trans_ values for *L. chinensis* (Figure 7d). Multiple regression analysis showed senesced leaf N concentration and NRE of the two species was primarily correlated with soil inorganic N concentration, while soil available P concentration was the main factor to be correlated to senesced leaf P concentration of two mixed species (Table 2). Soil moisture and available N:P ratio in the soil had a significant correlation relationship with PRE of *L. chinensis* and *M. sativa* (Table 2).

## 3. Discussion

### 3.1. Legume Proportions Influence Soil Water, N Fixation of M. Sativa, and N and P Availability in Soil

Since recent rainfall did not occur close to the sampling period, the soil water status reflected the consequence of long-term plant-soil feedback regulated by legume proportion. More legume likely caused greater plant water uptake and transpiration through the development of long root systems and development of a canopy, consequently reducing soil water [22,32]. In the current study, we did not identify a positive linear relationship between legume proportion and biological N_2_ fixation, since there was a decline in %Ndfa of *M. sativa* when its sowing proportion increased from 50% to 75% in grasslands. Similar results were found on a clover–grass mixture grassland in Europe [31]. The reduction of grass proportion and its facilitation effect on using fixed N_2_ from *M. sativa* was a potential mechanism for the decreasing %Ndfa in *M. sativa* [28,40]. Additionally, the decline in soil moisture appeared to limit N fixation [22]. The changes in N fixation by the legume will directly influence N availability of plants through symbiotic N transfer to soil and to neighboring grasses. As a consequence, a moderate proportion of legume in a mixture facilitated greater symbiotic N transfer to grass, and induced greater soil N availability compared with a high legume proportion. Our study found that high-L mixture decreased P availability in soil, suggesting an increased P limitation to plant growth under high legume proportion. 

### 3.2. Effects of Legume Proportions on Growth and Nutrient Uptake

Across different treatments, the observed density ratios of legumes in mixed grasslands were close to their initial sowing proportions, suggesting a strongly predetermined effect of initial plant establishment on community structure. Increasing legume proportions enhanced the total above-ground biomass of grass and alfalfa, primarily because the replacement of *L. chinensis* by the more competitive *M. sativa* [28]. Similarly, increasing the legume proportion significantly enhanced root biomass of the mixed grassland, as alfalfa commonly had greater individual root biomass than *L. chinensis*. In the current study, the low water availability may have driven more plant biomass allocation into the root, and, as a consequence, the root biomass varied between 93 to 112% of the total above-ground biomass in mixed grasslands with different legume proportions.

Changes of N and P availability in the soil are expected to alter the nutrient status of mature plants [4,14,15]. However, plants also maintain an internal nutrient status and adapt to environmental changes during long-term evolution, known as plant stoichiometric homeostasis [41]. In this study, the N and P concentrations and N:P ratios of green leaves for both species did not change with legume nutrient availability. These results likely indicate that *L. chinensis* and *M. sativa* have a high stoichiometric homeostasis during their growth. However, Lü et al. (2013) found that a more drastic change in soil N availability significantly altered green leaf N concentrations in *L. chinensis*. It is also possible that moderate changes in nutrient availability may limit plant responses in this study [13]. The N:P ratio in the green leaf can be a valid indicator of ecosystem nutrient limitations [18]. Plant growth is frequently considered to be P-limited when the N:P ratio is over 16 in green leaf [14]. In this study, the leaf N:P ratios ranged from 18 to 24 for both species, which indicated that this mixed grassland was P-limited. Increasing the legume proportion improved soil P acquisition, and induced a decline in soil P availability, but an increase of available N:P ratio in soil, which indicates that P limitations will increase with the increasing legume proportion in this mixed grassland.

### 3.3. Effects of Legume Proportions on N and P Resorption

In contrast with nutrients in green leaves, nutrients in senesced leaves of two mixed-species responded more sensitively to varied legume proportions in the current study. Accordingly, their nutrient resorption efficiencies showed a significant change with legume proportion. These results imply that plant nutrient re-use, rather than a change in nutrient uptake of plants, is the primary mechanism for plant adaptation to changes in nutrient availability in this mixed grassland [5]. It also indicated that varied legume proportions mainly influence nutrient concentrations of litter, rather than that of mature plants in mixed species. Other studies have suggested that N enrichment limited plant N resorption [13,14]. Our results showed that a moderate legume proportion reduced the leaf N resorption of two mixed-species more than a high legume proportion, which was inconsistent with our hypothesis. Two reasons may explain why this occurred. First, a moderate proportion of legume in the pasture caused greater symbiotic N fixation and potentially greater N transfer to *M. sativa* and *L. chinensis* growth, which reduced their dependence on soil mineral N. An increased supplementary N source may induce less N resorption under a moderate proportion than a high proportion of legume (Figure 4 and Figure 7) [39]. Moreover, under a moderate legume proportion, the greater symbiotic N_2_ fixation improved the N availability in soil, and reduced N resorption [13,14]. Consistent with our hypothesis, P resorption of two mixed plants tended to increase with increasing legume proportion. However, different environmental factors can drive changes in senesced leaf P concentration and P resorption efficiency. The soil P availability was the only factor that was related to senesced leaf P concentrations of the two mixed species, which suggests that the proportion of legume primarily regulated the extent of plant P resorption, and consequent litter P concentration, by altering soil P availability in this mixed grassland. The interesting result was that soil water, in place of soil P availability, was more closely correlated with P resorption efficiencies of two mixed-species following varied legume proportion, which highlights the importance of soil water in regulating plant P utilization in this mixed grassland. Less information is available to understand the influence of water availability on plant P resorption [14]. In this mixed grassland, decreased soil water content with increasing legume proportion likely enhanced P resorption efficiency of two mixed species, because the decline in soil moisture may indirectly reduce the senesced leaf P concentration and thus improve the completeness of plant P resorption by decreasing soil P availability. Based on this study, soil N availability for plants had no direct effect on plant P resorption. However, increasing soil N availability under moderate to high legume proportions likely enhanced P resorption of *M. sativa* via an increase of available N:P in soil (Table 2) [18].

In this study, the grass *L. chinensis* translocated 65 to 71% of leaf N and 63 to 75% of leaf P during senescence. Compared with *L. chinensis*, the *M. sativa* showed lower leaf N resorption (43 to 50%) and leaf P resorption (47 to 59%). The inter-specific differences in nutrient resorption resulted from the different leaf nutrient status between *L. chinensis* and *M. sativa* [42], as observed in a meta-analysis where resorption of leaf N and P decreased with increased nutrient status [10]. The N resorption of two mixed-species had a similar response to varied legume proportions (Table 2, Figure 6a), which suggests that change in the community structure had a similar influence on N resorption of *M. sativa* and *L. chinensis*. However, the P resorption of two mixed-species showed obviously different responses to changes in legume proportion. From low to moderate proportions of legume, P resorption of *L. chinensis* showed no significant change due to less changes in soil moisture and P availability. By contrast, P resorption efficiency of *M. sativa* increased significantly, resulting from increased available N:P ratio in the soil. On the contrary, from moderate to high legume proportions, a significant decline in soil moisture and P availability increased the P resorption by *L. chinensis.* However, P resorption efficiency of *M. sativa* did not change based on the available N:P ratio. These results imply that P utilization of mixed grassland is a more complex ecological process controlled by multiple factors compared with N utilization. The difference in P resorption by the two mixed species has driven significant change in the interspecific difference of litter P concentration and N:P ratios with varied legume proportion (Figure 5d,f), which may have a profound influence on litter decomposition and nutrient return [43].

In conclusion, changed legume proportion can alter soil water and nutrient availability of mixed communities, which significantly influences nutrient resorption and nutrient concentrations of senesced plants. Middle legume proportion decreases N resorption and thus increasing N concentration in senesced leaves of grass and legume species. P resorption generally is enhanced with increasing legume proportion in the two mixed species. As legume proportion changed, the different responses of P resorption by two mixed-species altered the interspecific difference for P concentration and N:P ratio of senesced leaves, which potentially influenced the further litter decomposition and nutrient return.

## 4. Materials and Methods

### 4.1. Study Site

The study site was located at the Changling Grassland Farming Research Station (E 123˚31′, N44˚33′) in Jilin province of China. This area is a semi-arid climate, with a mean annual temperature of 4.9 °C and annual precipitation of 364 mm from 2000–2010. The precipitation from May to September in 2010 is shown in Figure 1. The soil is classified as meadow chernozem soil. The mature vegetation is dominated by *Leymus chinensis (Trin.) Tzvel.*, a perennial warm-season grass. This experiment was conducted in the abandoned land that was converted from maize (*Zea mays*) planting in 2002. In 2006, when this experiment started, the main soil properties were as follows: 30% sand + 37% silt + 33% clay; pH 8.1; electrical conductivity 48.7 μs cm^−1^; bulk density 1.48 g cm^−3^; organic matter 16 g kg^−1^; and total N 1.1 g kg^−1^ at a depth of 0–30 cm.

### 4.2. Experiment design

Using a completely randomized block design, the restored grassland community was established in 2006 with 4 replicates. In each block, *Medicago sativa* L. and *L. chinensis* were sown into 3 mixes with 25% legume plant density (low legume proportion, Low-L), 50% legume plant density (middle legume proportion. Mid-L), and 75% legume plant density (high legume proportion, High-L), respectively. Each 3 × 3 m plot was separated by 0.5 m walkways. For each plot, we defined the initial target plant density (the combination of grass and legume) as 600 plant individuals m^−2^ which represents the average plant density of natural meadow communities in this region.

### 4.3. Experimental Set up

In July 2006, *Medicago sativa* L. and *L. chinensis* seeds were mixed in accordance with designated seedling density, but adjusted to ten percent above the actual seed germination rate and uniformly sown into plots with row spacings of 15 cm. Prior to sowing, grasses and weeds were eliminated in all plots by hand weeding. The seed coats of *M. sativa* seeds were scarified by soaking them in 98% H_2_SO_4_ for 30 min. No inoculation was applied at sowing, as the site had a history of lucerne cultivation between 2003-2004, and the previous experiment confirmed that the soil contained sufficient rhizobia to induce root nodulation [22]. To promote the successful seedling establishment of two mixed species, each plot was irrigated using 200 L water (equal to 20 mm precipitation) if no rainfall occurred during the latest 4 days in the month following sowing. In August 2006, plots were thinned to the designated initial plant densities, but no further density control was conducted after that. Plots were kept weed-free by hand-weeding between 2006 and 2007. After 2007, no weeding was conducted due to limited weeds presence which had a negligible impact on the growth of mixed plants. Between 2006 and 2010, the forage plots were not clipped or fertilized.

### 4.4. Samples Collection and Measurement

In early-September 2010 when total above-ground biomass of grass and alfalfa attained its peak value, a 1 × 1 m quadrat of vegetation was sampled from the center of each plot. First, plants were separated into *L. chinensis* and *M. sativa* species and counted, and then the shoot material of *L. chinensis* and *M. sativa* was separately cut at the soil surface. For each quadrat, we selected 20 shoots of *L. chinensis* and 20 shoots of *M. sativa* of similar size. Two fully expanded and intact green leaves (the third or fourth leaf from the top of the shoot) were collected at each selected *L. chinensis* shoot, and ten fully expanded and intact green leaves (located at 20–30 cm to the top of shoot) were collected at each selected *M. sativa* shoot. All the collected leaf samples and remaining plant samples were oven-dried (65 °C for 48 h) to determine their dry weight. The leaf samples and other intact shoot samples of two mixed-species finely ground. For leaf samples, total N concentration was determined using the Kjeldahl method [44], and total P concentration was analyzed by colorimetric analysis after persulfate oxidation [45]. The shoot samples of two mixed species were analyzed for total N concentration and ^15^N abundance using a continuous flow Isotope Ratio Mass Spectrometer (ThermoFisher MAT253, Waltham, MA, USA). In mid-October when plant leaves were fully senesced, 20 shoots of *L. chinensis* and 20 shoots of *M. sativa* were selected from the field, and the senesced leaves were sampled and analyzed as for green leaves.

In early-September, three soil cores from 0 to 40 cm soil depth were sampled from each plot using a soil corer with 5 cm diameter. These soil samples were sealed into three previously weighed aluminum containers in field, then their fresh weighed and dry weighed (oven-dried at 105 °C for 48 h) were measured in the laboratory, for calculating their gravimetric water content. Using a soil core sampler with 10 cm diameter, three more soil cores were sampled and bulked into a composite sample at the depth of 0–40 cm in each plot. Roots were washed free from the soil and the dry weight was determined after they were oven-dried at 65 °C for 48 h. An additional three soil samples at 0–40 cm soil depth were collected, and sieved to pass a 2 mm mesh to remove larger materials. Soil samples were analyzed for ammonium (salicylate method) and nitrate (cadmium reduction methodand) concentrations after being extracted with 50 mL of 2M KCl to using a Bran-Luebbe AA3 autoanalyzer (Bran and Luebbe, Hamburg, Germany). After extracting the soil with 0.5 M NaHCO_3_, soil available P concentration was determined using the molybdenum blue-ascorbic acid method [46].

### 4.5. Calculations

The proportion of N derived from the atmosphere (%Ndfa) in biomass of legumes was estimated using the following formula [47]:(1)%Ndfa=( δN15reference plant-δN15M.sativaδN15 reference plant-B),
where δ^15^N is the atom percent excess of ^15^N relative to atmospheric N. The subscript ‘reference plant’ represents *L. chinensis* growing in association with *M. sativa.* The ‘B’ is the δ^15^N from shoots of legumes that are fully dependent upon N_2_ fixation [47], which was cited from our previous study [22]. Total biological N_2_ fixation (Ndfa) was estimated based on %Ndfa, *M. sativa* shoot N concentration and shoot biomass.

The contribution of *M. sativa*-derived N (%N_trans_) to *L. chinensis* in mixtures was estimated according to the following formula [48]:(2)%Ntrans=100×( δN15 L.chinensis in monoculture-δN15 L.chinensis in mixtureδN15 L.chinensis in monoculture-B).

For *M. sativa* and *L. chinensis* in each plot, the nutrient pools of leaves were calculated according to the following formula:

(3)
Leaf nutrient pool=leaf nutrient concentration × total leaf mass)


Nitrogen resorption efficiency (NRE) or phosphorus resorption efficiency (PRE) was calculated based on leaf nutrient pools, which were calculated as:
(4)
NRE = (N pool_green_-N pool_senesced_) /N pool_green_× 100%

(5)
PRE = (P pool_green_-P pool_senesced_)/P pool_green_ × 100%

where N pool_senesced_ and P pool_senesced_ are the N or P pool of senesced leaves, and N pool_green_ and P pool_green_ are the N or P pool of green leaves in each plot, respectively.

### 4.6. Statistic Analysis

Prior to analysis, Shapiro–Wilk tests and Levene tests were used to examine the normality and equality of variance. A general linear model (GLM) was applied to examine the main and interactive effects of species identity and legume proportion on nutrient concentrations of green and senesced leaves, and leaf nutrient resorption efficiencies. One-way ANOVAs were used to analyze the effects of legume proportion on plant density, plant biomass, soil water, biological N fixation by *M. sativa* and %N_trans_ in *L. chinensis* and soil nutrient characteristics. The potential relationships between soil water and %Ndfa of *M. sativa,* and between legume-driven N and senesced leaf nutrients and nutrient resorption efficiencies in mixed-species were analyzed using linear regressions. Multiple stepwise regressions were used to reveal the correlations between soil water content, soil nutrient characteristics and leaf nutrient resorption. Duncan’s tests were performed to make meaningful comparisons among different legume proportions. A paired *t* test was used to analyze the inter-species difference under each legume proportion. The Significance level for all statistical tests was defined at *P* = 0.05. All data analysis was realized using the SPSS17.0 software (SPSS, Chicago, IL, USA).

## Figures and Tables

**Figure 1 plants-09-00292-f001:**
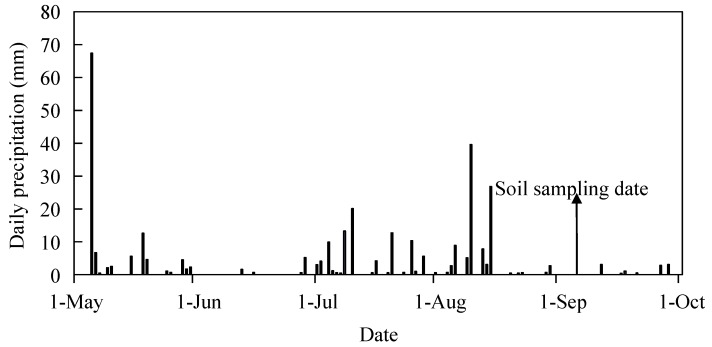
The daily precipitation from 1 May to 1 October in 2010. The arrow indicates soil samples were collected on 4 Septemer 2010.

**Figure 2 plants-09-00292-f002:**
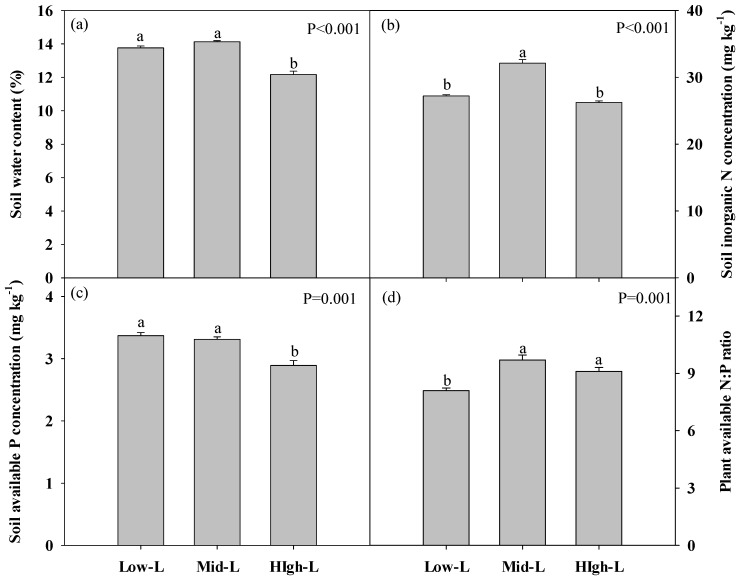
Soil water content (**a**), soil inorganic N concentration (**b**), soil available P concentration (**c**), and plant-available N:P ratio in soil (**d**) under different legume proportion, all measured for 0 to 40 cm depth. Vertical bars represent means ± SE for n = 4. Different letters above each bar indicate significant differences from each other according to Duncan’s multiple comparisons. Low-L, 25% legume; Mid-L, 50% legume; High-L, 75% legume.

**Figure 3 plants-09-00292-f003:**
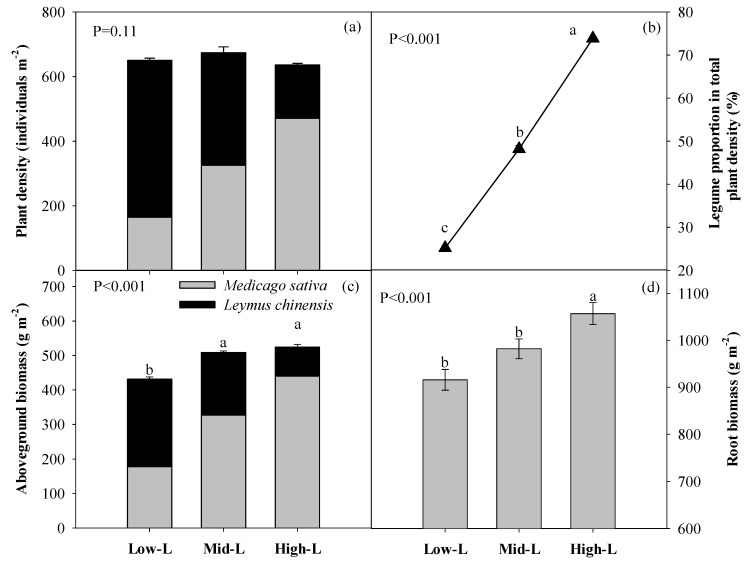
Plant density (**a**), legume proportion in total density (**b**), aboveground biomass (**c**), root biomass (0–40 cm soil depth, **d**) under different legume proportions. Error bars indicate ± SE (n = 4). Different letters indicate significant differences from each other according to Duncan’s multiple comparisons. Low-L, 25% legume; Mid-L, 50% legume; High-L, 75% legume.

**Figure 4 plants-09-00292-f004:**
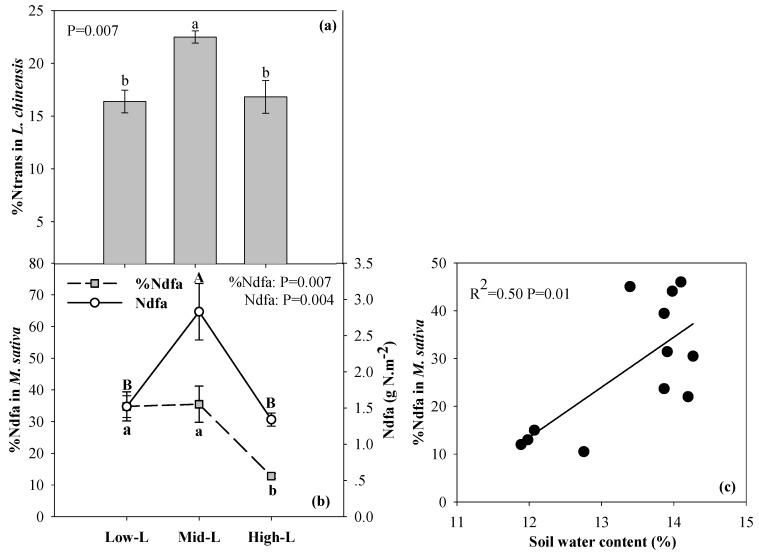
The contribution of *M. sativa*-derived N to *L. chinensis* (%N_trans_, **a**) and the proportion of N derived from the atmosphere (%Ndfa) and total biological N_2_ fixation (Ndfa) of *M. sativa* (**b**) under different legume proportion, and the relationships between %Ndfa in *M. sativa* and soil water content (**c**). Error bars indicate ± SE (n = 4). Different uppercase or lowercase indicates a significant difference between two legume proportions according to Duncan’s multiple comparisons. Low-L, 25% legume; Mid-L, 50% legume; High-L, 75% legume. The results in Figure 4b are taken from previously published research [39].

**Figure 5 plants-09-00292-f005:**
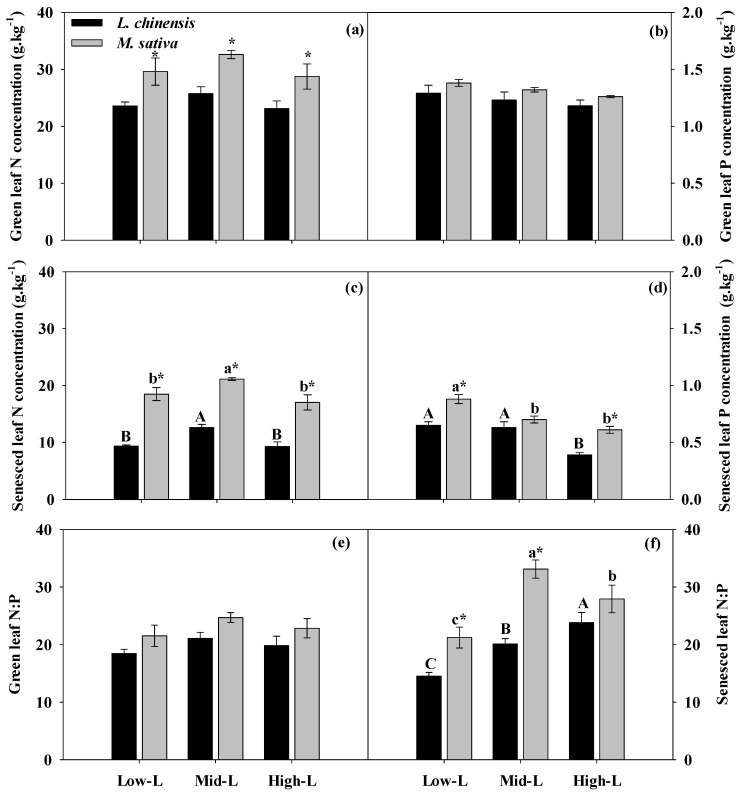
Green leaf N (**a**), green leaf P (**b**), senesced leaf N (**c**), senesced leaf P (**d**) concentration, green leaf N:P (**e**) and senesced leaf N:P (**f**) for *M. sativa* and *L. chinensis* under different legume proportions. Vertical bars show means ± SE (n = 4). Different uppercase or lowercase represent significant differences among legume proportions for *L chinensis* and *M. sativa,* respectively, according to Duncan’s multiple comparison tests. * indicates significant inter-species difference under each legume proportion according to paired *t* test. Low-L, 25% legume; Mid-L, 50% legume; High-L, 75% legume.

**Figure 6 plants-09-00292-f006:**
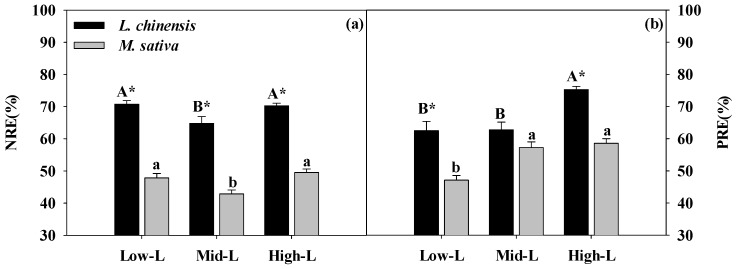
Nitrogen resorption efficiency (NRE, **a**), phosphorus resorption efficiency (PRE, **b**) for *M. sativa* and *L. chinensis* leaves under different legume proportion. Vertical bars show means ± SE (n = 4). Different uppercase or lowercase represent significant differences among legume proportions for *L chinensis* and *M. sativa,* respectively, according to Duncan’s multiple comparison tests. * indicates significant inter-species difference under each legume proportion according to paired *t* test. Low-L, 25% legume; Mid-L, 50% legume; High-L, 75% legume.

**Figure 7 plants-09-00292-f007:**
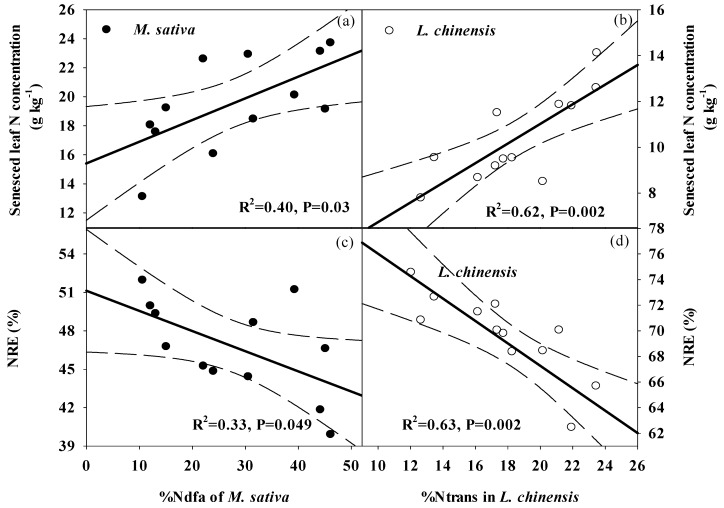
Relationships between %Ndfa and N concentration of senesced leaves (**a**), N resorption efficiency (NRE), (**c**) for *M. sativa,* and relationships between %N_trans_ and N concentration of senesced leaves (**b**), NRE (**d**) for *L. chinensis.* The dotted line indicates a 95% confidence interval.

**Table 1 plants-09-00292-t001:** Results (F and *p*-value) of two-way ANOVAs analysis on the effects of species (S), legume proportion (LP), and their interactions on nutrient concentrations in green and senesced leaves, and leaf nutrients resorption efficiency.

	S		LP		S × LP	
	F Value	*p* Value	F Value	*p* Value	F Value	*p* Value
Green leaf N	22.993	<0.001	2.371	0.122	0.082	0.921
Green leaf P	5.204	0.035	2.833	0.085	0.009	0.991
Green leaf N:P	8.243	0.010	2.223	0.137	0.028	0.972
Senesced leaf N	181.779	<0.001	18.680	<0.001	1.372	0.279
Senesced leaf P	41.867	<0.001	31.603	<0.001	3.779	0.043
Senesced leaf N:P	35.466	<0.001	17.658	<0.001	3.897	0.039
NRE	388.294	<0.001	12.219	<0.001	0.323	0.728
PRE	62.602	<0.001	19.651	<0.001	4.981	0.019

NRE, N resorption efficiency; PRE, *p* resorption efficiency.

**Table 2 plants-09-00292-t002:** The results of multiple regressions analysis with senesced leaf N concentration, N resorption efficiency (NRE), senesced leaf P concentration, or P resorption efficiency (PRE) as independent variable and soil water content, soil inorganic N concentration, soil available P concentration and plant available N:P in soil as dependent variables for *L. chinensis* and *M. sativa*.

	Response Variable	Regression Parameters for Each Dependent Variable						
		Intercept	Soil Water Content (%)	Soil Inorganic N (mg.kg^−1^)	Soil Available *p* (mg.kg^−1^)	Plant Available N:P in Soil	Overall R^2^	Overall *F* Value
*L. chinensis*	Senesced leaf N (g.kg^−1^)	−5.763	−0.180	0.567 **	0.112	−0.138	0.657	19.187 **
	NRE(%)	93.430 ***	0.156	−0.871 *	0.097	−0.117	0.383	6.198 *
	Senesced leaf P (g.kg^−1^)	−0.914 *	0.441	0.184	0.461 **	0.184	0.681	21.323 **
	PRE (%)	152.126 ***	−6.386 **	0.185	−0.190	0.151	0.630	17.039 **
*M. sativa*	Senesced leaf N (g.kg^−1^)	−5.871	0.278	0.892 **	0.184	−0.207	0.562	12.813 **
	NRE (%)	74.783 ***	−0.235	−0.983 **	−0.192	0.213	0.541	11.799 **
	Senesced leaf P (g.kg^−1^)	−0.537	−0.219	−0.416	0.398 **	−0.387	0.585	14.113 **
	PRE (%)	38.148	−2.697 *	−0.204	−0.191	5.824^**^	0.742	12.937 **

* *p* < 0.05; ** *p* < 0.01; *** *p* < 0.001.

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
