# Peer review of "Plant Nitrogen and Phosphorus Resorption in Response to Varied Legume Proportions in a Restored Grassland"

_plants, 2020, doi:10.3390/plants9030292_

Round 1

Reviewer 1 Report

Either authors really need to get permission from Fron Plant Sci to reproduce same figure which is already published in their previous work in Fron Plant Sci (Figure 4b). If there is no need, authors should cite the source of figure by citing their published work.

There are some minor English language issues and I do hope authors will address them during proof read. For instance,

L101-102: There was not any ......... to occur??

L192: .....that no a short??

Reviewer 2 Report

The authors have improved the manuscript based on the comments from me. This manuscript can be accepted from my side.

Author Response

This manuscript is a resubmission of an earlier submission. The following is a list of the peer review reports and author responses from that submission.

Round 1

Reviewer 1 Report

Please see attached comment's file.

Reviewer 2 Report

The authors investigated community biomass and characteristics of soil and plants N and P. The topic of this study is match with the journal scope. The experiment was carefully conducted and there are some new findings. I believe the accumulation of case study is highly important for comprehensive understanding especially for the field experiment. Therefore, I can recommend to publish this manuscript in Plants. I have some minor comment mainly editorial issues which should be improved before acceptance.

Fig. 1 and Table. 1 “the same below”

It is hard to understand the meaning of this phrase. May be, the meaning is “the abbreviations of experimental treatments are the same in the following figures (or tables)”. Please reconsider.

Figure 4 (b), (d)

“.5” -> “0.5”

Caption of Figure 5

L. chinensi sleaves” -> “L. chinensis leaves”

Table 2

Please reconsider the title of this table. This table seems to explain the results of multiple regression analysis. It is hard to understand because the sentence start from “Dependence of senesced leaf N concentration…”.

The numbers in the column of “Over all F value” are something wrong. Please correct them.

“4.3. Establishing of experiments”

Please consider “Set up of experiments” or Experimental set up”. I feel this is not “establishment”.
